# Current Status of Lymphangiogenesis: Molecular Mechanism, Immune Tolerance, and Application Prospect

**DOI:** 10.3390/cancers15041169

**Published:** 2023-02-11

**Authors:** Hongyang Deng, Jiaxing Zhang, Fahong Wu, Fengxian Wei, Wei Han, Xiaodong Xu, Youcheng Zhang

**Affiliations:** 1Hepatic-Biliary-Pancreatic Institute, Department of General Surgery, Lanzhou University Second Hospital, Lanzhou 730030, China; 2Key Laboratory of the Digestive System Tumors of Gansu Province, Lanzhou University Second Hospital, Lanzhou 730030, China

**Keywords:** lymphangiogenesis, immune tolerance, immunotherapy, meningeal lymphatic vessels, tumor metastasis, myocardial infarction

## Abstract

**Simple Summary:**

Lymphatic vessels are low-pressure, blind-ended tubular structures essential in maintaining tissue fluid homeostasis, immune cell transport, and lipid transport. More and more evidence showed that lymphangiogenesis might be closely related to the development of many diseases, and the intervention of lymphangiogenesis may be a new direction of disease treatment. This review aims to discuss the molecular mechanisms of lymphangiogenesis, the effect of lymphangiogenesis on tumor immune tolerance, the emerging role of meningeal lymphatics and cardiac lymphatics, and the promising applications of lymphangiogenesis in immunotherapy and bioengineering materials.

**Abstract:**

The lymphatic system is a channel for fluid transport and cell migration, but it has always been controversial in promoting and suppressing cancer. VEGFC/VEGFR3 signaling has long been recognized as a major molecular driver of lymphangiogenesis. However, many studies have shown that the neural network of lymphatic signaling is complex. Lymphatic vessels have been found to play an essential role in the immune regulation of tumor metastasis and cardiac repair. This review describes the effects of lipid metabolism, extracellular vesicles, and flow shear forces on lymphangiogenesis. Moreover, the pro-tumor immune tolerance function of lymphatic vessels is discussed, and the tasks of meningeal lymphatic vessels and cardiac lymphatic vessels in diseases are further discussed. Finally, the value of conversion therapy targeting the lymphatic system is introduced from the perspective of immunotherapy and pro-lymphatic biomaterials for lymphangiogenesis.

## 1. Introduction

The lymphatic system is a blind-ended vasculature network consisting of capillary lymphatic vessels, collecting lymphatic vessels, and secondary lymphatic organs such as lymph nodes (LNs). The lymphatic system plays an essential role in tissue fluid homeostasis, lipid absorption, immune surveillance, and transport of immune cells [1]. Lymphatic capillaries are composed of oak-leaf-like lymphatic endothelial cells (LECs). Adjacent LECs are loosely connected in a button-like manner, through which interstitial fluid and macromolecules can pass. These LECs express the chemokine CCL21 to guide the migration of dendritic cells (DCs) and other CCR7^+^ cells to them. LECs of the collecting lymphatic vessels are smooth and elongated and form tight, continuous chain connections. These LECs are covered with muscle cells that produce contractions to aid lymph flow [2]. In addition, they contain lymphatic valves that maintain unidirectional flow. The morphology of LECs in LNs is different. The LECs in the outer wall of the subcapsular sinus are similar to those in the lymphatic vessels and are closely organized to form a physiological barrier. On the contrary, LECs in the inner wall of the subcapsular sinus, cortex, and medulla are more like LECs of the primary lymphatic vessels, allowing immune cells to migrate through the endothelium between the lymphatic lumen and the LN parenchyma [3].

Recent single-cell sequencing of human LNs has revealed different subtypes of lymphatic endothelial cells with various functions, such as DC and lymphocyte recruitment, maintenance of T cell survival, antigen archiving [4], antigen presentation, and immune tolerance [5,6]. The functional diversity of LECs allows them to be either beneficial or harmful in the course of the disease. It can provide a pathway for antitumor immune cells to inhibit tumor progression, but more studies claim that tumor-associated lymphangiogenesis is associated with poor patient outcomes. Recent studies have shown that LECs in melanoma mainly inhibit antitumor immunity and promote immune tolerance and escape [7], but they can enhance the immunotherapeutic effect of melanoma [8]. The discovery of meningeal lymphatic vessels (MLVs) in recent years has shattered the perception that the central nervous system (CNS) is an immunologically privileged region [9]. Promoting MLV production can trigger antitumor-immune effects in brain tumors [10]. This seems to be a different outcome from melanoma but could also enhance immune/radiotherapy effects [11,12]. In addition, several studies have demonstrated in experimental models that promoting cardiac lymphangiogenesis after myocardial infarction (MI) reduces cardiac inflammation and fibrosis, thereby promoting cardiac functional recovery [13]. In conclusion, these studies suggest that lymphangiogenesis intervention may be a new strategy for the treatment of the disease (Figure 1). This review summarizes recent advances in lymphangiogenesis, including new findings on the signaling mechanism of lymphangiogenesis, the contribution of lymphangiogenesis to immune tolerance, the relationship between lymphangiogenesis and tumor metastasis, and its role in different diseases.

## 2. Lymphangiogenesis Signals

The lymphatic system plays a vital role in both physiological and pathological conditions. Exploring the molecular signaling mechanism of lymphangiogenesis may shed new light on the pathogenesis and prognosis of diseases. VEGFC/VEGFR3 and Prospero homeobox 1 (PROX1) are recognized as classical signals of lymphangiogenesis. Recent studies have found that these signals are related to lipid metabolism. Macrophages, extracellular vesicles (EVs), and mechanical signals are essential transducers of LEC proliferation. Multiple studies have focused on different aspects of the lymphangiogenic mechanism and may provide new therapeutic insights.

### 2.1. Classical Lymphangiogenesis-Related Signals

The vascular endothelial growth factor family (VEGF), which includes VEGFA, VEGFB, VEGFC, VEGFD, and placental growth factor (PLGF), can produce different functions by binding to other cell-surface tyrosine kinase receptors, including VEGFR1, VEGFR2, and VEGFR3. The binding effect of VEGFC and VEGFD with VEGFR3 is the main driving force of lymphangiogenesis [14]. The binding of VEGFA to VEGFR2 also promotes human lymphangiogenesis [15]. In addition, VEGFR2 can perform different functions by combining various molecules into different dimers, among which the dimers formed by the combination of VEGFR2 and VEGFR3 promote the migration and lymphangiogenesis of LECs [16,17]. When VEGFR2 is absent, lymphatic vessels are also observed to be dysplastic but still functional [15]. In summary, VEGFR3 is a primary affecting receptor for lymphangiogenesis, and VEGFR2 slightly affects lymphangiogenesis.

The process of lymphangiogenesis is divided mainly into four processes: proliferation, germination, migration, and the formation of vascular structures. In the central vein, the expression of the transcription factor PROX1 mediated by SOX18 and COUP-TFII in embryonic venous endothelial cells led to the formation of LEC progenitor cells [18]. Specifically, PROX1 activates VEGFR3 expression, and VEGFC-mediated activation of VEGFR3 signaling, in turn, maintains PROX1 expression [2]. At the same time, the maturation of pro-VEGFC in the embryonic stage depends on the participation of Adamts3 and CCBE1 [19]. In the adult stage, Adamts2/Adamts14 takes over from Adamts3 to process and activate pro-VEGFC into mature VEGFC [20]. Wong et al. found [21] that LEC development requires epigenetic regulation of fatty acid β-oxidation and lymphopoietic factors. A fatty acid β-oxidation rate-controlling enzyme (CpT1A) can be upregulated by PROX1 to promote fatty acid β-oxidation (FAO). FAO provides acetyl CoA (AcCoA) for histone acetic acid, which can be used by histone acetyltransferase p300 to acetylate histone H3K9ac at PROX1, thereby promoting VEGFR3 expression and ultimately promoting LEC proliferation and migration. Another source of AcCoA for this process is the regulation of autophagy by LECs. Autophagy in LECs is conducive to developing lipid droplets (LD), providing free fatty acids to mitochondria to promote FAO, which complements AcCoA [22]. Koltowska et al. discovered [23] Ddx21, a target molecule downstream of VEGFC/VEGFR3 signaling, which can regulate ribosome production, maintain the p53-dependent cell cycle of endothelial cells, and promote developmental lymphangiogenesis. LEC progenitor cells generally begin to bud after expressing PROX1 signals. They then express LEC differentiation markers such as podoplanin (GP38), and they gradually become arborized and lumenization. When fluid flow signals are detected, lymphatic valves develop and maintain unidirectional flow. The current understanding of the metabolic mechanisms is just beginning, and lipid metabolism has recently been found essential for regulating LEC differentiation (Figure 2). Adjusting dietary fatty acid intake may be an exciting area of research to regulate lymphatic vessel growth.

### 2.2. Macrophage-Associated Lymphangiogenesis Signals

Macrophages are one of the critical sources of VEGFC/VEGFD/VEGFR3. In the model of inflammation induced by lipopolysaccharide (LPS) in Gram-negative bacteria [24], LECs recruit macrophages to remodel lymphatic vessels by producing chemokines through LPS–Toll-like receptor 4 (TLR4)–NFKB signaling. The TLR4 signaling of macrophages enhances the expression of VEGFC and VEGFD to promote lymphangiogenesis. The P13K-Akt signal activated by the VEGFR3/VEGFC signal from macrophages promotes SOCS1 expression to inhibit the TLR4-NFkB signal, and it attenuates the release of inflammatory factors, thereby reducing the inflammatory response [25]. LECs and macrophages highly express VEGFC/VEGFD/VEGFR3 in the perfusion model of liver ischemia. VEGFR3, on the one hand, promotes lymphatic dilation around the portal vein to increase drainage; on the other hand, it can, in turn, drive macrophages to the repair phenotype [26]. In the myocardial infarction (MI) model, CD11b^+^ [27] and CD36^+^ [13] macrophages also express VEGFC to promote cardiac lymphangiogenesis and protect cardiac function. There is high expression of the podoplanin (PDPN) subtype in macrophages. Glycosylated PDPN combined with Galectin 8 (GAL8) can activate integrin-β1 and promote LEC adhesion and lymphangiogenesis, and this macrophage is closely related to lymphatic infiltration and lymphatic metastasis in breast cancer [28]. For other immune cells, Tregs can improve lymphedema and lymphatic drainage function in mouse models and may have a regulatory effect on lymphatic vessels [29]. However, Th2 and its secreted cytokines inhibit LECs-related transcription factors and LEC tubule formation [30]. In conclusion, macrophages are essential cells in tumor angiogenesis, and there is increasing evidence that they are also critical players in lymphangiogenesis [31]. The intervention of macrophage polarization at molecular and cellular levels to regulate tumor lymphangiogenesis opens a new horizon for the personalized treatment of cancer.

### 2.3. Other Lymphangiogenic Molecules

Recent studies have found that inactivating mutations in Angiopoietins 2 (Ang2) and Tie are associated with developmental disorders and loss of function in the human lymphatic system [32]. Korhonen et al. described [33] a novel mechanism by which Ang2/Tie activates the P13K/AKT pathway to inhibit Foxo1 and its downstream target genes in Ang2-related lymphangiogenesis. Akwii et al. found [34] that Ang2 can bypass Tie and use integrin-β1 to activate the downstream RhoA–formin axis to promote LEC migration and lymphangiogenesis.

The TGF-β pathway can also maintain the structure of lymphatic vessels and lymphatic homeostasis [35]. Zhu et al. demonstrated that TGFβR1 could mediate the lymphangiogenesis of bladder cancer through VEGFD signaling [36]. Lin et al. suggested [37] that TGFBIp could induce corneal lymphangiogenesis through the integrin–α5β1/FAK pathway. Pak’s in vivo and in vitro studies confirmed that TGF-β1 could promote the activation of VEGFC by Smad pathways in gastric cancer to promote lymphangiogenesis [38]. However, studies have shown [39,40,41] that TGF-β1 signaling can worsen lymphedema by impairing lymphangiogenesis during wound repair. Baik et al. confirmed [41] that TGF-β1 did not directly inhibit LEC but increased the infiltration of fibroblasts and Th1 cells and finally increased the hardness of the extracellular matrix (ECM), which inhibited the assembly of the lymphatic pipe network. These findings open up new ideas for the treatment of lymphedema.

Ephrin is another crucial molecular mechanism of lymphangiogenesis. Ephrin-b2, an Eph receptor tyrosine kinase transmembrane ligand, promotes vascular endothelial sprouting and angiogenesis and has extensive effects on cytoskeletal activity, cell adhesion, intercellular junctions, cell movement, and cell morphology [42]. Wang et al. found [43] that Ephrin-B2 promotes the internalization of VEGFR3 to enhance VEGFC/VEGFR3 signaling and is a vital regulator of this pathway. Blocking Ephrin-B2 dramatically reduced lymphangiogenesis and inhibited tumor growth in a mouse model [44]. Other studies have reported that EphrinB2–EphB4 signaling promotes the formation and maintenance of funnel valves in corneal lymphatic capillaries [45].

miRNAs are non-coding RNAs fewer than 200 nucleotides in length that bind to mRNA to inhibit transcription or translation [46]. They can be encased in EVs and play an important role in intercellular communication [47]. Recently, many miRNAs have been found [48] to regulate lymphangiogenesis. miRNAs mainly achieve this function by regulating mRNA that drive lymphangiogenesis [48,49,50], such as VEGFC/VEGFR3, PROX1, FOXO1, etc., while they are also regulated by upper long non-coding RNAs [36]. Thus, miRNA regulation of lymphangiogenesis is multidimensional and needs further exploration.

Exploring the molecular mechanisms and signaling pathways that regulate lymphangiogenesis is still ongoing (Table 1). With further research in this field, new findings may provide a reliable reference for diagnosis, treatment, and prognosis assessment of lymphangial-related diseases.

### 2.4. Effect of EVs on Lymphangiogenesis

EVs are bilayer lipid particles released by eukaryotic cells that carry bioactive molecules and play an important role in cell-to-cell communication. Several studies have confirmed that EVs containing tumor-released lymphangiogenic factors are essential mechanisms of tumor lymphangiogenesis. Chen et al. found [51] that EV-transported ELNAT1 could mediate the SUMO-dependent UBC9/SOX18 signaling axis and further promote lymphangiogenesis in bladder cancer cells. Exosomes from bladder cancer cells can also carry LNMAT2, thus promoting LECs to express PROX1 to drive lymphangiogenesis [52]. The loss of DUSP2 in pancreatic ductal adenocarcinoma (PDAC) promotes the processing of VEGFC, which is subsequently transported by EVs to the tumor microenvironment (TME), leading to lymphangiogenesis [53]. EVs secreted by PDAC cells with KRAS mutation can carry SUMOylated hnRNPA1 into LECs and upregulate the expression of PROX1 to promote lymphangiogenesis [54]. Endometriotic cells also promote lymphangiogenesis and immune cell infiltration by secreting EVs carrying VEGFC, thereby exacerbating inflammation [55]. Melanoma-derived extracellular vesicles carry NGFR into LECs and LN macrophages, activate NF-kB/VEGFR3 signaling in LECs, and promote LEC proliferation [56].

In conclusion, EVs are essential mediators of tumor progression, which target lymphangiogenesis and may also be involved in tumor lymphatic metastasis. Table 2 reviews studies in which lymphangiogenic factors carried by EVs promote lymphangiogenesis. In the future, more evidence is needed to demonstrate the role of EVs in promoting tumor lymphatic metastasis.

### 2.5. Mechanical Signals Regulate Lymphatic Vessels and Lymphatic Valves

The microenvironment of LECs regulated by mechanical signals plays an essential role in shaping lymphatic physiology, especially in driving the generation of lymphatic valves to ensure one-way drainage of lymphatic fluid. Transcription factors such as FOXC2, GATA2, Orai1, Piezo1, and the Wnt signaling pathway can drive lymphatic valve development in response to oscillatory shear stress (OSS) (Figure 3). The disturbance of fluid flow can cause LEC proliferation and cell death [60], and the same phenomenon is also observed in vascular endothelial cells [61]. Resistance to this mechanical force is essential for developing the lymphatic system.

FOXC2 is a critical transcription factor that stabilizes and maintains the function of collecting lymphatic vessels. FOXC2 is highly expressed in collecting lymphatic vessels, especially endothelial cells, from lymphatic valves [60]. After sensing mechanical signals, FOXC2 mediates valvular lymphatic formation via Cx37–Cn/NFATc1 signaling [62]. It can block the proliferative effect of Hippo/YAP1/TAZ signals collected from lymphatic vessels, ensure the quiescence of LECs, and increase the anti-disturbance of LECs by maintaining the intercellular connections and the cytoskeleton under the induced liquid shear force [60]. Loss of FOXC2 causes morphological changes in lymphatic valves, leading to severe functional impairment [60]. Hernandez et al. found [63] that FOXP2, a downstream target molecule of FOXC2, is highly expressed in collecting lymphatic vessels. It can not only participate in the indirect activation of valve generation by the FOXC2/NFATc1 pathway but can also directly activate valve generation by FOXC2 regulation. Yang et al. demonstrated [64] that after sensing mechanical signals, VE-cadherin can regulate β-catenin and AKT signaling in the nucleus, activate PROX1 and FOXC2 expression to promote lymphoid valve development, and also maintain valve function by blocking TAZ signaling [65].

GATA2, a member of the zinc finger family of transcription factors, was initially found to be highly expressed in lymphatic valves, and it is also a critical molecule that mediates the development of lymphoid valve development [66]. Subsequent studies showed [67] that GATA2 could cooperate with Lmo2 to regulate NRP2 transcription, thereby regulating VEGF-mediated lymphangiogenesis. Betterman et al. identified [68] FAT4, a downstream effector target of GATA2, which, upon receiving mechanical fluid signals, drives LECs to polarize towards a phenotype of cell rearrangement and cell migration that favors valve formation [69]; at the same time, FAT4, Adamts3, and CCBE1 function in the same signaling pathway to promote VEGFC processing and maturation. Recent studies have also found that GATA2 transcription is regulated by the hardness of the ECM at which LECs are located [70].

FOXO1 is a negative regulator of vascular development [71]. The phosphorylation of AKT induced by sensing the bidirectional flow shear force can inactivate FOXO1, downregulate the transcription factor PRDM1, relieve the transcriptional inhibition of Cx37 and FOXC2, and induce valvular formation [72].

Mutations in PIEZO1, a fluid-flow mechanical signal receptor for cationic calcium channels, cause systemic lymphatic dysplasia. It can activate another calcium channel (Orai1) to promote calcium influx. Then calmodulin (CaM) binds to PROX1/Klf2 to form a transcriptional complex, which binds to the promoters of DTX1 and DTX3L to initiate their transcription. Finally, DTX1 and DTX3L block the Notch pathway to promote lymphatic sprouting. However, how Piezo1 regulates Orai1 is still unknown because both PIEZO1 and PIEZO1 are membrane channel proteins. Choi et al. speculated that the mechanism might be physical [73].

The Wnt/β-catenin pathway is also a key regulator of lymphatic and valve development, and it has recently been reported [74,75] to sense OSS signaling to regulate FOXC2, GATA2, and PROX1 activation. Integrin is a transmembrane receptor essential for intracellular and extracellular signaling, and integrin-β1 is involved mainly in the development and generation of lymphatic vessels. Integrin-β1 is a target molecule that senses the mechanical signal of lymphatic expansion caused by LEC stretching due to interstitial fluid accumulation, which increases VEGFR3 phosphorylation and promotes LEC proliferation [76]. Urner et al. elucidated [77] a novel mechanism by which ILK negatively regulates integrin-β1/VEGFR3 signaling and prevents lymphatic overgrowth, whereas mechanical stretching signaling can block ILK function. Podoplanin expressed by TAM and GAL8 secreted by LECs can activate integrin-β1 in a glycosylated manner and promote the infiltration of LECs [28]. In vitro, LEC tubule formation experiments of HA hydrogels confirmed that LECs sense matrix hardness through YAP/TAZ mechanoreceptors and that the soft matrix promotes the expression of VEGFC/VEGFR3 and MMP14 to promote LEC migration and LEC tubule formation [78]. 

Shear force and matrix stiffness appear to be the primary mechanical signals whose abnormal conduction leads to lymphatic and lymphatic valve dysfunction, and they are an important cause of lymphedema. Intervention in these mechanical signals may be more promising than massage and compression bandages for lymphedema.

## 3. Modulating the Effect of Lymphatic Vessels on Tumor Immunity

Lymphatic vessels play an important role in immune regulation by coordinating the transport of antigens and immune cells from peripheral tissues to the collection of lymphatic vessels and LNs. LECs can regulate immune cell migration and immune effects through various secreted factors, the most famous of which is the lipid sphingosine one phosphate (S1P). The lymphatic system responds to tumor antigens or exogenous antigens to activate adaptive immunity and has an immune tolerance mechanism to help immune escape. Currently, an increasing number of studies are focusing on the effect of tumor immune tolerance on the lymphatic system. 

### 3.1. S1P

S1P is a G-protein-coupled receptor, mainly derived from LECs in the lymphatic system, that controls the migration of immune cells from S1P-low to S1P-high environments [79]. Mature T cells in the thymus also need the help of the S1PR1 signal to leave the thymus [80] and then sense the S1P signal gradient through S1PR1 and S1PR4 to cross the LECs. Moreover, T cells also need LECs to express S1PR2 [81]. However, autophagy in LECs can reduce S1P production, inhibit T cell migration, reduce T-cell-associated autoimmunity [82], and enhance naive T cells’ survival and mitochondrial function [83]. Baeyens et al. found [84] that inflammatory monocytes could supply S1P in LNs through CD69, and a high level of S1P would prolong the residence time of T cells in LNs. Loss of S1P leads to accumulation and ectopia of natural killer (NK) cells, resulting in reduced efficiency against Salmonella [85]. S1P regulates endothelial cell spread, maturation, stability, and barrier integrity [86]. Since S1P can regulate lymphatic permeability, it may play a role in a lymphatic invasion during tumor lymphatic metastasis [87].

### 3.2. Mechanisms of Immune Tolerance in the Lymphatic System

Increasing evidence supports the immunosuppressive role of lymphatic vessels, which can reduce inflammation and promote tumor immune escape, especially in melanoma models.

Cutaneous malignant melanoma is one of the most aggressive malignant tumors. It progresses rapidly and readily metastasizes through the lymphatic system [88]. The density of lymphatic vessels in human melanoma has been reported to be closely correlated with T cell infiltration and immunosuppressive molecules such as nitric oxide synthase (iNOS) and 2, 3-dioxygenase (IDO) expression, suggesting that melanoma-associated lymphatic vessels activate both antitumor and antitumor immune effects [89]. Although enhanced tumor-associated lymphangiogenesis may increase the presentation of tumor antigens to the specific immune system, it appears deleterious in melanoma. Several studies have confirmed that the net benefit of VEGFC release from melanoma cells and tumor-associated macrophages (TAM) to induce lymphangiogenesis is to promote LN metastasis of melanoma [90,91,92,93], which is considered a marker of a poor prognosis of melanoma [94,95,96,97].

In the murine B16 melanoma model, VEGFC was shown to contribute to tumor immune tolerance by promoting naive T cell loss in sentinel LNs and cross-presenting tumor antigens by LECs, leading to CD8^+^ T cell dysfunction and apoptosis [7]. Activation of CD8^+^ T cells requires the presentation of tumor-associated antigen (TAA) by antigen-presenting cells (APC) carrying major histocompatibility class I complexes (MHC-I) [98]. Similarly, LECs can perform APC presentations by cross-presenting TAA. This cross-presentation effect is similar to that of liver sinusoidal endothelial cells (LSEC), which are the first cells to respond to food antigens. Their cross-presentation helps the immune system to absolve these foreign antigens, facilitating protein processing in the liver [99,100]. LECs with high expression of MHC-I and PD-L1 cross-presented activated CD8^+^ T cells, which carried more PD-1, CTLA4, and CD80 than activated DCs. These CD8^+^ T cells secrete only small amounts of IFN-γ and IL-2 and express low activation markers such as CD25, CD44, and CD69. These cells are depleted and dysfunctional early on and cannot be reversed by IL-2 [101]. Similarly, in the B16 mouse melanoma model, IFN-γ promoted the expression of MHC-II in LECs. MHC-II^+^ LECs presented TAA, which increased the number of Treg cells and decreased the number of effector T cells. The number of Treg cells was positively correlated with the density of lymphatic vessels [102]. LECs, fibroblastic reticular cells (FRCs), and blood endothelial cells (BECs) belong to the lymph node stromal cells (LNSCs) subgroup. The expression of LNSCs is only partially regulated by IFN-γ and depends on EVs secreted by DCs. Moreover, acquired pMHC-II can promote the dysfunction and apoptosis of CD4^+^ T cells after the presentation to these cells [103]. In conclusion, although LECs can present TAA in the melanoma model, the activation effect is quite different from that of “professional APCs,” and it is always immunosuppressive.

LECs express various peripheral tissue antigens (PTAs), which present melanocyte-specific protein tyrosine kinases to CD8^+^ T cells, resulting in the loss of CD8^+^ T cells [104]. At the same time, LECs can deliver these PTAs to DCs to induce tolerance of CD4^+^ T cells, and MHC-II molecules in LECs can mediate tolerance of CD8^+^ T cells through LAG-3 [105]. Contact between DCs and LECs induces ICAM-1-mediated contact inhibition, which inhibits the maturation of DCs and the ability to stimulate T cell proliferation [106].

IFN-γ signaling in lymphatic vessels is also one of the crucial mechanisms of immune suppression and immune escape, which can promote the expression of PD-L1 in LECs through the JAK/STAT pathway to inhibit T cell accumulation [107]. Encapsulated miR-1468-5p in cervical cancer exosomes can also target the JAK/STAT3 pathway activated by HMBOX1 in LECs, promoting lymphangiogenesis, high expression of lymphatic PD-L1, and destroying T cell immunity [108]. In colorectal cancer, the VEGFC/VEGFR3 pathway induces the proliferation of LECs and recruitment of macrophages, but VEGFR3 induces the polarization of TAM to the M2 type, which, together with LECs, inhibits the proliferation of CD4^+^ T cells and CD8^+^ T cells [109].

In conclusion, LEC-mediated immune tolerance is mainly through the following mechanisms: 1. inhibition of DC maturation, 2. secretion of immunosuppressive factors (IDO, iNOS, and TGFβ signaling molecules), 3. expression of immune checkpoints (PD-L1, CTLA4, LAG-3, etc.), 4. downregulation of T cell costimulatory molecules such as CD28, CD27, 4-1BB, and OX40 and inhibition of T cell function by inhibiting IL-2 [110], and 5. carrying MHC-I/II or presenting PTA antigens (Figure 4).

## 4. The Relationship between Lymphatic Vessels and Tumor Metastasis

Metastasis is the leading cause of cancer-related death, and the relationship between lymphatic vessels and tumor progression has been the subject of much research. Early theories only supported lymphangiogenesis as the biological pathway of tumor metastasis [111]. However, later evidence showed that tumor lymphangiogenesis is an immunosuppressive effect [112], promoting tumor lymphatic colonization and providing a suitable microenvironment for distant metastases.

### 4.1. Relationship between Lymphatic Vessels and Lymphatic Metastasis of Tumors

Many cancers, such as melanoma, breast, cervical, and gastric cancer, can metastasize through the lymphatic system. Due to the bidirectional immunomodulatory function of tumor-associated lymphatic vessels, its correlation with lymphatic metastasis has always been controversial [112,113]. As described above, the immunosuppressive/tolerant microenvironment facilitated by LECs makes tumor-associated lymphatics a pathway for tumor cells to colonize LNs rather than a transport pathway for leukocytes. With the progression of the tumor, lymphangiogenesis is gradually increased, which, coupled with the remodeling and dilation of the collecting lymphatic vessels, increases the flow velocity of the vessels and dramatically increases the drainage and transportation capacity of the lymphatic vessels [114]. In addition, lymphatic vessels express large amounts of CCL21, which provides migration guidance for tumor cells expressing the CCL21 receptor CCR7 and drives tumor cells to migrate into the lymphatic system [115]. Cancer stem cells are a class of tumor cells with self-renewal and differentiation abilities associated with relapse, metastasis, and drug resistance [116]. The recently discovered lymphatic stem cell niche provides a protective resting environment for tumor stem cells, which persist in LNs even after resection of the primary tumor [117,118]. The accumulation of evidence supports the idea that lymphatic vessels contribute to tumor metastasis. Mouse melanoma models lacking lymphatic vessels at all also lack antitumor immune responses.

Interestingly, lung metastasis is reduced in this setting. In the initial stage of the tumor, the lymphatic system may show more of an antitumor effect than pro-tumor effect and only begin to show a full pro-tumor effect when the tumor progresses to a particular stage. Precise temporal and spatial control of lymphangiogenesis, rather than blindly blocking lymphangiogenesis, may be an effective strategy to prevent lymphatic metastasis of tumors.

### 4.2. Relationship between Lymphatic Vessels and Distant Tumor Metastases

Although most tumors with distant metastases are preceded by LN invasion, the relationship between trans-lymphatic and distant metastasis has been controversial [119]. There are two hypotheses about the relationship between the lymphatic system and distant metastases. One hypothesis thought that tumors colonizing LNs would shift to a phenotype favoring distant metastasis and then spread to other organs. Another hypothesis is that lymphatic metastasis is not associated with distant metastases. The density of lung metastases is associated with a poor prognosis in patients with melanoma. In a mouse model, overexpression of VEGFC in the lung promoted lymphatic infiltration and lung metastasis of melanoma, with more metastasis to other distant organs [120]. Naxerova et al. analyzed [121] 213 biopsy samples from 17 colorectal cancer patients and found that 65% of lymphatic and distant metastases had tumor cells of different subtypes, and the remaining 35% had common subtypes. In a study of 1934 patients with melanoma, the presence or absence of sentinel LN dissection did not improve survival [122]. The loss of the LEC barrier switch S1P/SPNS2 resulted in circulating lymphocytopenia, accumulation of effector T and NK cells in the lung, and reduced melanoma metastasis [123].

Several recent studies support the idea that the lymphatic system drives distant metastases. The traditional view is that metastatic tumor cells enter the lymphatic system and eventually migrate from the thoracic duct to the subclavian vein and systemic circulation [124]. Brown et al. found [125] that collecting lymphatic vessels can transport tumor cells to the floor of the subcapsular sinus of the LN, where tumor cells enter the LN stroma and then enter blood circulation through high endothelial venules. Therefore, the high endothelial venules (HEVs) of LNs are the outlet for murine breast cancer cells to enter systemic circulation before lung colonization, which is more efficient than direct lung metastases of primary tumors, indicating that lymphatic vascular channels are at least part of the route of tumor cell metastasis. Invasion of tumor cells appears to be enhanced after entering the lymphatic system [125]. This phenomenon has also been demonstrated in mouse models of squamous cell carcinoma and melanoma [126]. This increased invasiveness may be related to the immune tolerance effects induced by LN colonization of tumor cells, which were subsequently discovered by Reticker et al. [119]. They believed that after tumor cells colonized LNs, MHC-I expression was upregulated to avoid NK cell killing, and PD-L1 was upregulated to inhibit T cell function in response to IFN signals and to induce Treg differentiation, thus establishing a tolerant microenvironment and facilitating distant metastasis of this type of tumor cell [119].

In conclusion, distant metastatic tumor cells and lymphatic metastatic tumor cells may not be identical isoforms. At least the lymphatic system provides a partial outlet for the distant metastasis of tumor cells, and lymphatic-system-mediated immunosuppression also provides a metastatic microenvironment for the distant metastasis of tumor cells. Therefore, the lymphatic system may predict distant metastasis and be a therapeutic target for cancer.

## 5. The Role of the Lymphatic System in CNS Diseases

In the past, lymphatic drainage of the brain was thought to require the transport of lymphocytes and cerebrospinal fluid to cervical LNs employing the crib’s lamina and nasal mucosa [127]. However, it used to be considered an immunologically privileged site because the brain is rich in microglia, lacks other immune cells, and has no lymphatic system like peripheral tissue [128]. This immunity privilege means that tumors can grow unchecked [129], and there is no immune rejection of grafts [130]. Medawar [130] believes this immune privilege stems from a lack of the blood-brain barrier and lymphatic drainage system. More and more studies have disproved the theory of the immune privileged zone in the CNS because the CNS immune system, although different from the peripheral immune system, also has functional lymphatic vessels [131].

In 2012, Illiff discovered [132] the glymphatic system, an AQP4-dependent cerebrospinal fluid and interstitial fluid exchange system that is functionally homologous to the peripheral lymphatic system that can remove peripheral waste products from nerve cells. Mascagni and his colleagues discovered the presence of lymphatic vessels in the meninges as early as 1787. Recent studies have also shown that the MLVs located in the dorsal and sub-basal part of the skull are critical pathways for the central nervous system to exocytose macromolecules and transport immune cells to the cervical lymph node (CLN) [9,133,134]. Similarly, human-like MLVs have been found in the CNS of zebrafish, which is sensitive to VEGFC signaling and could be used as a novel model to study MLVs [135].

The lymphatic system may be a double-edged sword for the CNS, as it plays a vital role in immune cell trafficking, antigen presentation, induction of antitumor immune responses, fluid drainage, and increased immunotherapy sensitivity while exacerbating pathological neuroinflammatory processes. High expression of VEGFC and PROX1 always predicts poor prognosis in peripheral organ tumors. However, their tumor-associated lymphangiogenesis downregulates the invasiveness of pediatric medulloblastoma (MB) [136] and increases the immune surveillance of glioma [10]. Ahn et al. found [134] that aging is related to the decline of MLV function. The dysfunction of MLVs and the glymphatic system can lead to amyloid deposition, impaired learning, and cognitive dysfunction in young adult mice. Enhancing MLV function in elderly mice can improve their cognitive function. Therefore, impaired MLV function may be one of the causes of cognitive dysfunction and Alzheimer’s disease (AD) in aged mice [137]. Chen et al. described [138] the mechanism of red blood cell drainage from MLVs to the CLN after subarachnoid hemorrhage (SAH). Inhibition of MLV production aggravated the neurological symptoms of SAH, indicating that MLVs may be an essential way to remove red blood cells from SAH. In experimental intracerebral hemorrhage (ICH) models, late ICH shows enhanced MLV production, and inhibition of MLV production reduces hematoma clearance. In contrast, increased MLV production can help clear the hematoma, improve behavioral symptoms, and reduce brain residual red blood cells, iron deposition, neuronal necrosis, and astrocyte activation [139]. Late hepatic encephalopathy (HE) is a severe neurological complication in patients with cirrhosis. Hsu and colleagues observed that increased MLV production promoted MLV drainage in HE, reduced NF-kB signal transduction and microglial phagocytosis, improved neuroinflammation in the brain, and alleviated motor dysfunction in HE model rats [140].

After the stroke, the VEGFC/VEGFR3 signal-dependent proliferation of LECs occurs in CLN, and LECs, in turn, drive the activation of pro-inflammatory macrophages, thus increasing neuroinflammatory-related brain injury. Inhibiting the VEGFC/VEGFR3 pathway or CLN resection can alleviate this brain injury [141]. The proliferation of lymphatic vessels near the cribriform plate during autoimmune encephalomyelitis (EAE) helps drain cerebrospinal fluid, cells, and antigens, leading to DC migration and T cell proliferation, thereby exacerbating neuroinflammation. However, EAE does not induce MLVs de novo, indicating that the lymphatic vessels of the CNS are functionally heterogeneous [142]. Hsu and colleagues [143] performed single-cell RNA sequencing of MLVs near the lamina crib ride during neuroinflammation and found upregulation of antigen-presenting genes. This lymphatic vessel is rich in CD11C^+^ and CD4^+^ T cells for antigen presentation, thus forming an immunomodulatory niche, which may be one of the reasons why MLVs exacerbate neuroinflammation. In conclusion, in-depth insights into the anatomy and function of the lymphatic system of the CNS suggest that MLVs may be a novel therapeutic target for CNS diseases such as brain tumors, ICH, and neuroinflammation.

## 6. The Role of Cardiac Lymphatics

The mammalian heart also has an extensive network of lymphatic capillaries. The puritan collecting lymphatic vessels drain lymphatic fluid to the periaortic and paratracheal mediastinal lymph nodes (MLNs) [144] and expel metabolic waste as the heart contracts and relaxes. Recently, cardiac lymphatics have been discovered to have therapeutic potential for cardiovascular diseases.

Initially, it was thought that the only origin of LECs was embryonic veins. In contrast, Klotz found that LECs of cardiac lymphatics have two origins: the venous endothelium and the yolk sac [145]. Furthermore, they suggest that cardiac lymphangiogenesis after the ischemic injury is promoted, similar to lymphangiogenesis, which promotes inflammation resolution after skin infection. This effect may also help resolve myocardial inflammation and improve cardiac function after MI [145]. MI can also cause dysfunction of the lymphatic vessels around the heart scar, poor fluid drainage, and edema. Delivery of VEGFC genes by albumin-alginate by Henri et al. promoted the regeneration of cardiac lymphatic vessels and saved the harmful remodeling of the collecting lymphatic vessels [146].

Furthermore, Vieira et al. demonstrated that cardiac lymphatic angiogenesis after MI could transport pro-inflammatory macrophages to MLNs to alleviate inflammation, a process based on LYVE-1 [147]. Recently, Glinton found that efferocytosis can regulate CD36^+^ macrophages’ secretion of VEGFC to promote lymphangiogenesis and inhibit macrophages’ over-secretion of pro-inflammatory cytokines, thus improving cardiac function [13,148]. The expression of VEGFC was also observed to be upregulated in regenerated coronary endothelial cells. It activated the signaling axis, thus promoting the proliferation of coronary endothelial cells in zebrafish heart injury models [149]. This suggests that VEGFC can promote cardiac regeneration and repair by the proliferation of LECs and coronary endothelial cells. In rat cardiac ischemia-reperfusion models, VEGFC targets VEGFR2 and activates Akt signaling, thus promoting Bax expression, blocking mitochondrial membrane translocation, protecting cardiomyocytes from H2O2-mediated apoptosis, and showing a dose-dependent reduction in infarct size for VEGFC [150].

Similarly, the absence of VEGFC/VEGFR3 signaling in mouse models of cardiac hypertrophy increases cardiac hypertrophy and dysfunction, while VEGFC transmission improves hypertrophy and delays the development of centripetal heart failure [151]. The bioactive peptide apelin has been implicated in tumor lymphangiogenesis and promotes lymphatic metastasis [152]. However, Tatin et al. found [152] that apelin regulates LEC secretion of S1P after MI to maintain the integrity of the cardiac LEC barrier and is beneficial to cardiac homeostasis, which is also an exciting strategy for treating ischemic diseases by intervening in lymphangiogenesis. After the adrenal medulla hormone (AM/Adm) drives MI, the expression of Cx43 connexin of LECs promotes the coupling of LEC gap junctions and reduces the dilatation and edema of the cardiac lymphatic system, thus improving cardiac function after MI [153]. In conclusion, the protective role of cardiac lymphatic vessels in cardiovascular disease manifests itself primarily in the following ways: 1. protein exudation and cholesterol transport, 2. inflammation and immune response, 3. liquid equilibrium [154], 4. anti-cardiomyocyte apoptosis, and 5. promote coronary endothelial proliferation.

Cardiac lymphatic vessels have shown good cardioprotective potential in experimental models, and the VEGFC gene/protein delivery system targeting the heart may be a new therapeutic approach for cardiovascular diseases. Zhang et al. used SAP hydrogels to deliver VEGFC and lymphatic endothelial progenitor cells (LEPCs) to the myocardial tissue, effectively alleviating cardiac edema, myocardial fibrosis, and the inflammatory environment in MI [155]. Qiao et al. constructed a HepNP–VEGFC complex intravenous delivery system using VEGFC and negatively charged heparin polysaccharide nanoparticles (HepNP). In acute MI, HepNP–VEGFC therapy has been shown to eliminate edema, reduce scarification, and improve cardiac function. It is even more effective when administered with a fractional VEGFC/VEGFA [156]. Houssari et al. found that cardiac-infiltrated T cells could secrete IFN-γ to inhibit the formation of cardiac lymphatic vessels.

Meanwhile, amplifying VEGFC with adenoviral vectors promotes therapeutic lymphangiogenesis, which accelerates the regression of cardiac inflammation after MI, reduces the level of left ventricular T cell and pro-inflammatory macrophage infiltration, delays scar remodeling, and reduces cardiac dysfunction after MI [157]. Therapeutic lymphangiogenesis, which improves cardiac function after MI by reducing myocardial edema, inflammation, and fibrosis, has shown potential in experimental models. The prognostic correlation of inflammatory cardiac lymphangiogenesis and its cardioprotective effect in patients with heart disease needs to be further explored in the future [158].

## 7. Effect of Lymphangiogenesis on Immunotherapy

Although tumor lymphangiogenesis is primarily immunotolerant, the transport function of immune cells and antigens in lymphatic vessels is still necessary to activate adaptive immunity. Multiple immunotherapies have been developed to target tumors, but they do not benefit all patients. Recently, multiple studies have shown that lymphangiogenesis enhances immunotherapy responses, which may be a promising sensitizer for patients who do not respond well to immunotherapy.

In the absence of dermal lymphatic vessels, the implantation of B16 melanoma in mice stimulated only a tiny amount of immune cell infiltration and cytokines, which was also demonstrated in the analysis of the correlation between human lymphatic markers and the level of immune cell infiltration. Furthermore, the OVA vaccine (ovalbumin) could not activate CD8^+^ T cells, indicating that the antitumor immune response depends on tumor lymphatic vessels [159]. VEGFC has recently been shown to activate CCL21/CCR7 signaling in a mouse model to promote the activation and recruitment of naive T cells to tumors, enhancing the efficacy of adoptive T cell therapy (ATT), DC vaccines, and CpG TLR9 ligand CpG [8]. Chemokine receptor 7 (CCR7) was expressed in naive T cells, regulatory T cells, memory T cells, mature DCs, and B cells. As a receptor for CCL21 and CCL19, CCR7 induces the directed movement of lymphocytes and regulates immune and tolerance responses [160]. Maria et al. [161] developed an immunotherapeutic vaccine to induce lymphangiogenesis using genetically modified lethally irradiated tumor cells to overexpress VEGFC. The vaccine induced a persistent specific T cell immune response in a mouse melanoma model, causing delayed tumor growth.

Glioblastoma (GBM) is adults’ most lethal primary brain malignant tumor. The lymphocytes infiltrating into the TME of GBM are mostly depleted dysfunctional T cells, immune-suppressing TAM, and functionally suppressed NK cells. Therefore, the TME of GBM lacks T cell infiltration and shows no survival benefit against immunotherapy such as immune checkpoint inhibitors, CAR T cells, and DC vaccines [162,163].

The CNS TME lacks tumor-associated lymphatic vessels compared to peripheral tissue tumors and thus has limited immune surveillance capacity. Meningeal lymphatic angiogenesis facilitated by increased expression of VEGFC promotes lymphatic drainage, tumor antigen presentation, and immune surveillance. It can induce an intense and persistent T-cell-dependent antitumor immune response against GBM that reverses immune escape. The mouse model treated with targeted VEGFC therapy in combination with immunotherapy showed significant survival benefits compared to those treated with anti-immune checkpoint inhibitors alone [10]. Murine glioma and melanoma cells with brain metastases can specifically induce remodeling of the dorsal MLV but not in the basal MLV and nasal lymphatic vessel.

Furthermore, the dorsal MLV is a crucial channel for intratumor fluid, tumor cells, and DCs to transport to CLN. VEGFC-stimulated MLV production increased DC drainage, which increased the number of CD8^+^ T cells and CD8^+^Ki67^+^ T cells and inhibited the activation of CD4^+^Foxp3^+^ Tregs cells. Meanwhile, the effect of chemotactic T cell recruitment of CCL21/CCR7 can enhance the effect of anti-PD-1/CTLA4 immunotherapy. In conclusion, enhancing MLV production may be an effective method for immunotherapy for brain tumors [11]. Radiotherapy is the first-line therapy for GBM. It has recently been found that its sensitivity is also dependent on the lymphatic system of the CNS due to its ability to modulate the immune environment of GBM. Radiation therapy combined with VEGFC activates CCL21, promotes the transport of DCs, and increases the number of CD8^+^Ki67^+^ T cells, Treg cells, and overall CD8^+^ T cells in CLN, thus showing a higher sensitivity to GBM and metastatic brain tumors [12].

Without the cooperation of effector immune cells, immunotherapy is futile. In conclusion, lymphatic vessels enhance immunotherapy efficacy in four ways (increasing the immune-activating effect of DC, increasing the chemotaxis of immune cells to the lymphatic system, inhibiting immunosuppressive cell activation, and increasing immune effector cell activation), which precisely compensates for the immune tolerance phenotypes of the TME. This makes it a promising immunotherapy partner.

## 8. Tissue-Engineered Biomaterial for Lymphangiogenesis

Lymphangiogenesis has shown excellent potential in preclinical studies of tumor immunotherapy, cardiovascular disease, lymphedema, and anti-brain tumor. Biomaterials can generate temporal and spatial regulation of lymphangiogenesis, which may be a novel therapeutic strategy to deal with the progression of multiple diseases.

Hydrogels are a particular class of biomaterials with solid-like characteristics consisting of cross-linked polymers that mimic the properties of the ECM to improve cell adhesion, survival, and function and can be used to deliver small molecules, proteins, endothelial cells, and stem cells. The therapeutic effects of hydrogels are mainly achieved through the following three ways: controlling drug release, supporting or guiding tissue growth, and carrying foreign cells into the native tissue. It has some advantages in promoting therapeutic lymphangiogenesis due to its spatiotemporal controllability [164]. Table 3 summarizes recent research on tissue engineering materials for lymphangiogenesis. Due to its adjustability, the poly(ethylene glycol) (PEG) hydrogel system has been proven to promote lymphatic vessel germination in vitro and in vivo [165].

Hyaluronic acid (HA) is a non-sulfated glycosaminoglycan that regulates lymphangiogenesis [166]. HA hydrogels can bind with LYVE-1 homologs, which is an excellent substrate that can mimic the lymphatic neophyte environment [167] and can also mediate the transport of DCs [168] to block the recruitment of neutrophils [169]. The injection of HA hydrogels in the MI mouse model alleviated scar formation and collagen deposition, demonstrating that this material has a particular application potential [170]. HA can also be chemically modified to enhance its functional diversity. For example, a promising approach is to modify HA with norbornene groups [171]. BioBridge [172] is a nanofibrous collagen scaffold that increases the density of the lymphatic collecting duct in a porcine model. This scaffold is promising for the treatment of lymphedema.

Hydrogels made of various biological materials, such as collagen, fibrin, and alginate, have been used in tissue engineering for lymphangiogenesis. Polyacrylate (PEGDA) hydrogels have been widely used in bone tissue [173] engineering and angiogenesis [174]. PEGDA hydrogels have good mechanical plasticity compared with other hydrogels and are a potential matrix to promote lymphatic angiogenesis. In conclusion, tissue-engineered biomaterials that promote the generation of the lymphatic system may be an effective platform to study the molecular mechanism and function of the lymphatic system and may also be a new therapeutic strategy for lymphedema and cardiovascular diseases.

LEPCs: lymphatic endothelial progenitor cells; SAP: self-assembling peptide.

**Table 3 cancers-15-01169-t003:** Examples of tissue-engineered materials used to promote lymphangiogenesis.

Year/Authors	Materials	Effect	Reference
2022/Hooks et al.	PEG-3MAL hydrogels	Promote the sprouting of collected lymphatic vessels sprouting	[165]
2014/Marino et al.	Collagen type I and fibrin hydrogel	Form lymphatic capillaries in vitro within 21 days	[175]
2007/Helm et al.	VEGF-fibrin-collagen hydrogel	Promote lymphangiogenesis	[176]
2016/Hadamitzky et al.	Aligned nano fibrillar collagen scaffolds (BioBridge)	Alleviate the porcine lymphedema model	[172]
2017/Campbell et al.	Alginate hydrogels release of VEGFC/VEGFD	Therapeutic lymphangiogenesis	[177]
2014/Li et al.	PEI-alginate nanoparticles deliver VEGFR3-siRNA	Suppress tumor lymphangiogenesis and lymphatic metastasis	[178]
2021/Chávez et al.	Fibrin-collagen scaffolds of SynHA cyanobacteria	Promote lymphangiogenesis in dermal regeneration scaffold	[179]
2011/Hwang et al.	VEGFC hydrogel	Promote lymphangiogenesis in a mouse model	[180]
2019/Zhang et al.	Combined delivery of LEPCs and VEGFC with SAP	Promote cardiac lymphangiogenesis and repair of the infarcted myocardium	[155]
2020/Qiao et al.	Hep@VEGFC delivery system	Reduce scar formation and improve cardiac function	[156]
2020/Houssari et al.	VEGFCadeno-associated viral gene delivery of VEGFC	Accelerate the resolution of cardiac inflammation after MI	[157]

## 9. Conclusions and Prospects

In conclusion, lymphatic vessels are the crossroads of tumor metastasis, inflammation, and immunity. Their complex functions, especially their dual roles in immunity, prevent them from being amplified or suppressed blindly. Exploring the lymphangiogenesis signals and understanding the heterogeneity of lymphangiogenesis function in different signal generations, anatomical locations, and diseases may be an essential step for accurately regulating lymphangiogenesis function in a favorable direction. The recently discovered lymphatic immune tolerance mechanism is the umbrella of tumor metastasis, and we believe it may exert more antitumor effects in the early stage of the tumor. Understanding when and under what conditions tumor-associated lymphatic vessels begin to change from beneficial to harmful effects may be an important challenge in this field. This immune tolerance mechanism also shows good prospects in antagonizing organ transplantation rejection [181]. Meanwhile, the enhanced immunotherapeutic effect of lymphatic system amplification in brain tumors and melanoma may be a new hope for immunotherapy-insensitive patients. In addition, the beneficial role of lymphatic vessels in preclinical models of cardiovascular disease opens up new therapeutic strategies for saving cardiac function. The lymphatic system is an emerging field with great potential for disease treatment, but it still has a long way to go.

## Figures and Tables

**Figure 1 cancers-15-01169-f001:**
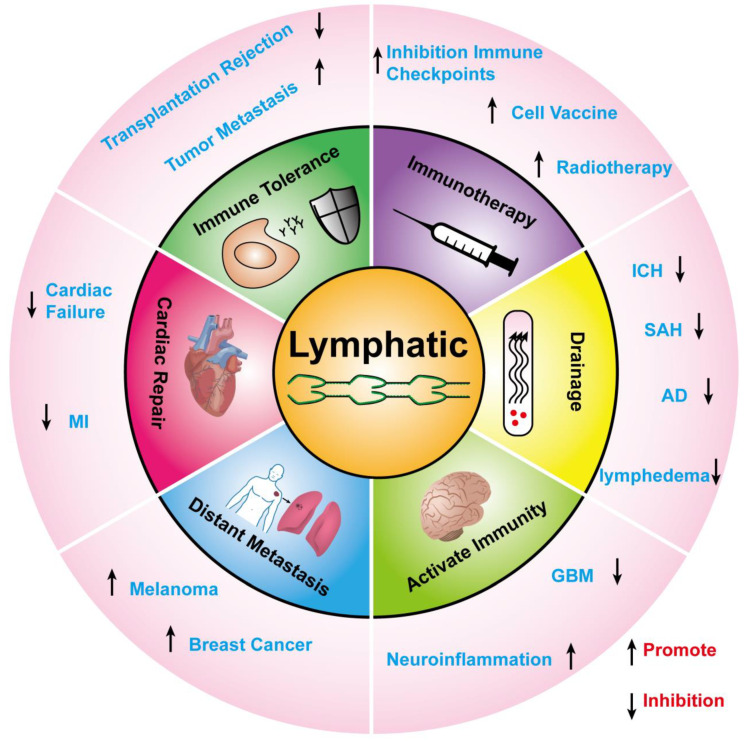
The function of lymphatic system.

**Figure 2 cancers-15-01169-f002:**
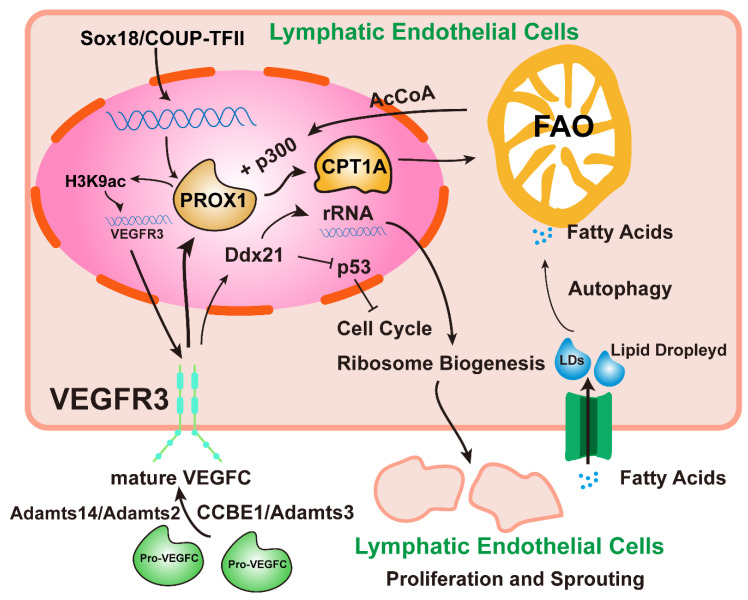
Relationship between VEGFC- and PROX1-related signaling pathways and lipid metabolism. SOX18 and COUP-TFII promote PROX1 transcription and the differentiation of embryonic venous endothelial cells into LECs. Fatty acids enter LECs and aggregate into lipid droplets that are then transported to mitochondria by autophagy to provide the free fatty acids required for FAO. AcCoA is produced by fatty acid oxidation, and acetyltransferase p300 acetylates histone H3K9ac of PROX1 to promote VEGFR3 expression. Prox1 promotes CpT1A-dependent fatty acid β-oxidation to further increase AcCoA production. In addition to interacting with PROX1, VEGFC/VEGFR3 signaling can also regulate the expression of ribosomal RNA through Ddx21 and inhibit the positive regulation of the cell cycle by p53, thus promoting the proliferation of LECs. Endogenous pro-VEGFC must be cleaved by CCBE1/Adamts3 and Adamts14/Adamts2 to become mature VEGFC, which can bind to VEGFR3 and regulate PROX1 expression.

**Figure 3 cancers-15-01169-f003:**
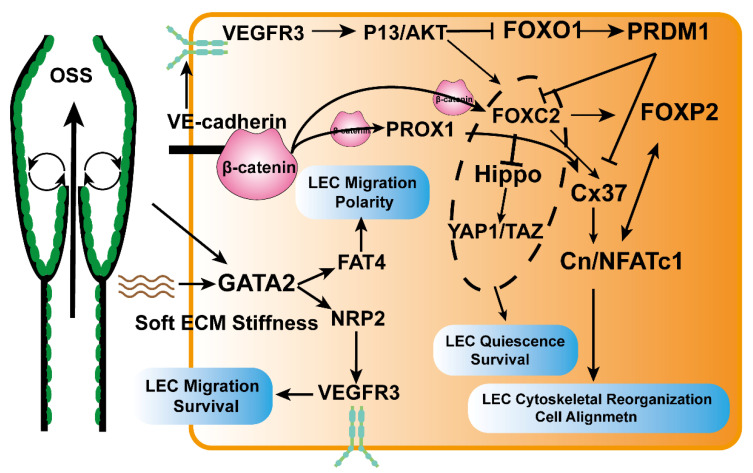
Molecular mechanism of the effect of mechanical signals on LECs. Transcription factors GATA2, FOXC2, and FOXO1 are important target molecules in LECs regulated by mechanical signaling. In response to OSS signaling, VE-cadherin can bind to β-catenin to drive FOXC2 and PROX1 transcription and can phosphorylate VEGFR3 to activate the P13/AKT pathway. AKT phosphorylation can inhibit the transcription of FOXO1 and induce the expression of FOXC2. FOXO1 inhibits the activity of FOXC2 and Cx37 by regulating PRDM1 and inhibits valvular lymphatic production. FOXC2 inhibits the Hippo pathway and downstream YAP1/TAZ to aid in the quiescence and survival of LECs. In addition, FOXC2 cooperates with PROX1 and FOXP2 to regulate Cx37 and Cn/NFATc1 to control LEC cytoskeleton remodeling and cell alignment in response to OSS. OSS and soft ECM stiffness can activate transcription of GATA2. GATA2 mediates LEC migration and polarity via FAT4 on the one hand and migration and survival via NRP2/VEGFR3 on the other.

**Figure 4 cancers-15-01169-f004:**
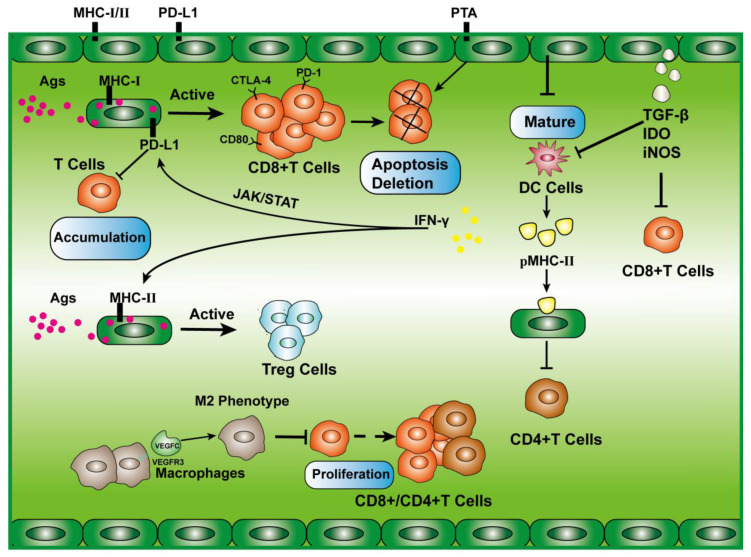
Mechanisms of immune tolerance mediated by the lymphatic system. IFN-γ signaling promotes the expression of MHC-I molecules and PD-L1 by LECs, which present immune checkpoints such as CTLA-4, PD-1, and CD80 expressed by CD8^+^ T cells activated by foreign antigens. These CD8^+^ T cells undergo apoptosis. LECs expressing PTA also result in the loss of CD8^+^ T cells. LECs secrete immunosuppressive molecules such as TGF-β, IDO, and INOS to inhibit CD8^+^ T cells and DCs. LECs can inhibit DC maturation. Presentation of antigens by LECs expressing MHC-II activates Treg cells and inhibits effector T cells. The expression of VEGFR3 by macrophages and VEGFC can promote the polarization of macrophages to the M2 type, and M2-type macrophages can inhibit the proliferation of effector T cells. LNSCs obtain pMHC-II from DCs to inhibit CD4^+^ T function.

**Table 1 cancers-15-01169-t001:** Examples of molecules and signaling pathways that promote lymphangiogenesis.

The Pathways of Lymphangiogenesis	Reference
VEGFC or VEGFD/VEGFR3	[14]
VEGFA/VEGF2	[15]
VEGFR2–VEGFR3 dimer	[16,17]
SOX18 and COUP-TFII/PROX1	[18]
Adamts3 and CCBE1/pro-VEGFC/mature VEGFC	[19]
Adamts2/Adamts14/pro-VEGFC/mature VEGFC	[20]
PROX1/FAO/VEGFR3	[21]
VEGFR3/Ddx21/p53	[23]
LEC autophagy/LD/FAO/PROX1/VEGFR3	[22]
LPS/TLR4/VEGFRC and VEGFD	[24]
PDPN/GAL8/integrin-β1	[28]
Ang 2/Tie/PI 3 K/VEGFR3	[33]
Ang2/integrin-β1/RhoA	[34]
circEHBP 1/TGF-β/SMAD 3/VEGFD	[36]
TGFBIp/integrin-α5β1/FAK	[37]
TGF-β1/Smad/VEGFC	[38]
Ephrin-B2/VEGFR3	[43,44]

**Table 2 cancers-15-01169-t002:** Examples of molecules carried by EVs that promote lymphangiogenesis in various diseases.

Year/Authors	Disease	The Molecules of EVs Contain	Effector Target Molecule	Reference
2021/García et al.	Melanom	NGFR	NF-kB/VEGFR3	[56]
2020/Li et al.	Endometriosis	VEGFC	VEGFR3	[55]
2020/Wang et al.	PDAC	VEGFC	VEGFR3	[53]
2021/Luo et al.	KRAS mutant PDAC	hnRNPA1	PROX1	[57]
2021/Chen et al.	BCa	lncRNA ELNAT1	SOX18	[51]
2020/Chen et al.	BCa	lncRNA LNMAT2	PROX1	[52]
2019/Zhou et al.	CSCC	miR-221-3p	Inhibit 9VASH1	[58]
2019/Yang et al.	HCC	miR-296	EAG1/VEGFA	[47]
2019/Wang et al.	OSCC	Laminin-γ2	Integrin-α3	[59]

PDAC: pancreatic ductal adenocarcinoma; BCa: bladder cancer; CSCC: cervical squamous cell carcinoma; HCC: hepatocellular carcinoma; OSCC: oral squamous cell carcinoma.

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
