# Peer review of "Current Status of Lymphangiogenesis: Molecular Mechanism, Immune Tolerance, and Application Prospect"

_cancers, 2023, doi:10.3390/cancers15041169_

Round 1

Reviewer 1 Report

This is a systematic review. Deng et al. reviewed the molecular mechanisms related to lymphangiogenesis, its influence on tumor immune tolerance, meningeal lymphangiogenesis, cardiac lymphangiogenesis, and its value in transforming immunotherapy and tissue engineering materials. The topic is new and interesting. The manuscript is written in a fluent and understandable way. The figures are of good quality and were well illustrated. The article is logical and well structured. The references are comprehensive and are of high quality published in recent years.

I have a few concerns about this review:

Major Comments

1. The current title is a bit long. The title needs to be refined.

2. There are a few errors in the formatting of nouns. For example, “smad” in 1.3 should be written as “Smad”.

3. "Effect of lymphangiogenesis on immunotherapy" might be more appropriate for a sixth subtitle.

4. [118] Reference marks should be added after Brown at al. in 3.2. There seems to be no reference in an entire sentence

5. There are some abbreviations that are not summarized in the abbreviations list.

6. “lymph nodes” should be written as “LNs” from the second occurrence

7. Moderate English changes are required.

Author Response

Dear Reviewer 1:

We appreciate you for your precious time in reviewing our paper and providing valuable comments. It was your valuable and insightful comments that led to possible improvements in the current version. We have carefully considered the comments and tried our best to address every one of them. We hope the manuscript after careful revisions meet your high standards. We welcome further constructive comments if any.

Below we provide the point-by-point responses.

Major comment 1: The current title is a bit long. The title needs to be refined.

Response 1: Thank you for your kind reminders. We've adjusted the title.

Major comment 2:There are a few errors in the formatting of nouns. For example, “smad” in 1.3 should be written as “Smad”

Response 2: Thank you for your careful reading of this manuscript. We have corrected these errors.

Major comment 3:"Effect of lymphangiogenesis on immunotherapy" might be more appropriate for a sixth subtitle.

Response 3: Thank you very much for your valuable suggestion. We have amended the sixth subtitle as requested.

Major comment 4:[118] Reference marks should be added after Brown at al. in 3.2. There seems to be no reference in an entire sentence

Response 4: Thank you very much for your valuable suggestion. After the modification, this is 4.2, where we added the reference.

Major comment 5:There are some abbreviations that are not summarized in the abbreviations list.

Response 5: Thank you for your careful reading of this manuscript. We complete the abbreviations list.

Major comment 6:“lymph nodes” should be written as “LNs” from the second occurrence

Response 6: Thank you for your careful reading of this manuscript. We've replaced it with abbreviations.

Major comment 7:Moderate English changes are required.

Response 7: Thank you for your careful reading of this manuscript. We carefully adjusted the grammar of the manuscript and corrected grammatical errors wherever possible.

Reviewer 2 Report

In this review current status of lymphangiogenesis: have lack of consistency and missing many latest published review article references on lymphangiogenesis. Many similar reviews were written on lymphangiogenesis. In my opinion, this review needs to be examined and updated with the most recent publications. However, any known role of micro-RNAs in lymphangiogenesis mechanisms of tumor resistance to therapy should be mentioned.

Author Response

Dear Reviewer 2:

We appreciate you for your precious time in reviewing our paper and providing valuable comments. It was your valuable and insightful comments that led to possible improvements in the current version. We have carefully considered the comments and tried our best to address every one of them. We hope the manuscript after careful revisions meet your high standards. We welcome further constructive comments if any.

Below we provide the point-by-point responses.

Major comment 1: this review needs to be examined and updated with the most recent publications.

Response 1: Thank you for your suggestion. We have replaced some of the reference literature with more recent ones, but have retained some of the classic important reference literature.

Major comment 2:However, any known role of micro-RNAs in lymphangiogenesis mechanisms of tumor resistance to therapy should be mentioned.

Response 2: Thank you very much for your suggestion. We found that miRNA plays an important role in regulating mRNA related to lymphangiogenesis. We discussed the role of miRNA in lymphangiogenesis in Part 2.3.

Reviewer 3 Report

This review is of interest and well-written. I propose some modifications in order to improve the manuscript:

- I propose to add a table concerning the different lymphangiogenic molecules with the signaling pathways and the references of the articles.

- I propose to edit the review in order to modify some mistakes. For example, p11 line 7: "a large number of lymphangiogenesis";  p11 line 16: lns; page 12 line 6: "not necessarily isoforms"; page 14 lines 1 to 3: change the formulation to remove the  ";" ; the same for page 14, end of the first paragraph. Check all the text.

- Define what are "cold tumors" page 14 paragraph 7, for readers who are not familiar with this notion.

- rearranfge the abbreviations in alphabetic order for an easier reading.

Author Response

Dear Reviewer 3:

We appreciate you for your precious time in reviewing our paper and providing valuable comments. It was your valuable and insightful comments that led to possible improvements in the current version. We have carefully considered the comments and tried our best to address every one of them. We hope the manuscript after careful revisions meet your high standards. We welcome further constructive comments if any.

Below we provide the point-by-point responses.

Major comment 1:  I propose to add a table concerning the different lymphangiogenic molecules with the signaling pathways and the references of the articles.

Response 1: Thank you for your kind suggestion. We added this table at the end of 2.3 as you suggested (Table1).

Major comment 2: I propose to edit the review in order to modify some mistakes. For example, p11 line 7: "a large number of lymphangiogenesis";  p11 line 16: lns; page 12 line 6: "not necessarily isoforms"; page 14 lines 1 to 3: change the formulation to remove the  ";" ; the same for page 14, end of the first paragraph. Check all the text.

Response 2: Thank you for your careful reading of this manuscript. We have corrected the above error.

Major comment 3:Define what are "cold tumors" page 14 paragraph 7, for readers who are not familiar with this notion

Response 3: Thank you very much for your valuable suggestion. We have replaced "cold tumors" with the meaning of "cold tumors" in the manuscript.

Major comment 4:rearranfge the abbreviations in alphabetic order for an easier reading

Response 4: Thank you very much for your valuable suggestion. We have rearranged the abbreviations in alphabetical order.

Reviewer 4 Report

In this manuscript the authors made a good review of interesting aspects of both physiological and pathological lymphangiogenesis.

Overall the manuscript is well detailed and I only have few comments :

The paragraph at the start of page 3 feels out of place. It does not make a clear link between part 1. and 2. and as all the different parts have their own introduction it is useless so i would recommend to delete it.

Part 2.1 does not take into account the role of VEGFR2 in lymphangiogenesis. A short part on its implication in lymphangiogenic signaling based notably on the papers from Lena Claesson-Welsh and/or Agnès Noel should be added. In addition, this part only mentions Adamts3 as a protein required for VEGC maturation while Dupont et al demonstrate that both Adamts2 and Adamts14 are also implicated in pro-VEGFC activation (PMID: 35316211).

In Part 2.3, TGF-b pathway is presented as only beneficial for lymphatic vessels while in lymphatic-related diseases such as lymphedema, TGF-b has been clearly established  to be detrimental for lymphatic function (PMID: 18849330, 21056998 and 35652284). Both roles should be discussed in this part.

In part 6, the role of Apelin in cardiac lymphatics is not mentioned despite being published (PMID: 28614788). Given the known implication of apelin in restoring heart function after MI and in lymphangiogenesis (notably in cancers), it is worth adding this protein in this part.

Author Response

Dear Reviewer 4:

We appreciate you for your precious time in reviewing our paper and providing valuable comments. It was your valuable and insightful comments that led to possible improvements in the current version. We have carefully considered the comments and tried our best to address every one of them. We hope the manuscript after careful revisions meet your high standards. We welcome further constructive comments if any.

Below we provide the point-by-point responses.

Major comment 1: The paragraph at the start of page 3 feels out of place. It does not make a clear link between part 1. and 2. and as all the different parts have their own introduction it is useless so i would recommend to delete it.

Response 1: Thank you for your suggestion. We deleted this part and added a few sentences to the previous paragraph.

Major comment 2:Part 2.1 does not take into account the role of VEGFR2 in lymphangiogenesis. A short part on its implication in lymphangiogenic signaling based notably on the papers from Lena Claesson-Welsh and/or Agnès Noel should be added. In addition, this part only mentions Adamts3 as a protein required for VEGC maturation while Dupont et al demonstrate that both Adamts2 and Adamts14 are also implicated in pro-VEGFC activation (PMID: 35316211).

Response 2: Thank you very much for your reminding. After looking at Lena Claesson-Welsh and/or Agnes Noel's articles, we found that dimers produced by VEGFR2 do promote lymphangiogenesis, and we added these. We found that Adamts2 and Adamts14 promote Pro-VEGFC maturation in adulthood, and we added this content in part 2.1 and figure 2.

Major comment 3:In Part 2.3, TGF-b pathway is presented as only beneficial for lymphatic vessels while in lymphatic-related diseases such as lymphedema, TGF-b has been clearly established to be detrimental for lymphatic function (PMID: 18849330, 21056998 and 35652284). Both roles should be discussed in this part.

Response 3: Thank you very much for your valuable reminder. We found a negative effect of TFG-β on the lymphatic system and discussed them.

Major comment 4:In part 6, the role of Apelin in cardiac lymphatics is not mentioned despite being published (PMID: 28614788). Given the known implication of apelin in restoring heart function after MI and in lymphangiogenesis (notably in cancers), it is worth adding this protein in this part.

Response 4: Thank you very much for your valuable suggestion. We added the role of apelin in Part 6 Fourth paragraph.

Round 2

Reviewer 2 Report

The manuscript is well organized and made significant changes. I would like to recommend to publish in Cancers